# Older but Stronger: Development of Platinum-Based Antitumor Agents and Research Advances in Tumor Immunity

**Jianing Liu** [1,†]**, Yi Cao** [1,†]**, Bin Hu** [1,†]**, Tao Li** [1,†]**, Wei Zhang** [1]**, Zhongze Zhang** [1]**, Jinhua Gao** [1]**, Hanjing Niu** [1]**, Tengli Ding** [1]**, Jinzhong Wu** [1]**, Yutong Chen** [1]**, Pengfei Zhang** [1]**, Ruijuan Ma** [1]**, Shihao Su** [1]**, Chaojie Wang** [2]**, Peng George Wang** [3]**, Jing Ma** [1,*] **and Songqiang Xie** [1,*]

1. School of Pharmacy, Academy for Advanced Interdisciplinary Studies, Institute of Chemical Biology, Henan University, Kaifeng 475004, China
2. The Key Laboratory of Natural Medicine and Immuno-Engineering, Henan University, Kaifeng 475004, China
3. School of Medicine, The Southern University of Science and Technology, Shenzhen 518055, China
* Correspondence: majing.1988.ok@163.com (J.M.); xiesq@henu.edu.cn (S.X.)
† These authors contributed equally to this work.

**Abstract:** Platinum (Pt) drugs have developed rapidly in clinical applications because of their broad and highly effective antitumor effects. In recent years, with the rapid development of immunotherapy, Pt-based antitumor agents have gained new challenges and opportunities. Since the discovery of their pharmacological effects in immunotherapy and tumor microenvironment regulation, research into Pt drugs has progressed to multi-ligand and multi-functional Pt precursors and their own shortcomings have been further highlighted. With the development of antitumor immunotherapy and the rise of combination therapy, the development of Pt-based drugs has started to move in the direction of multi-targeting, nanocarrier modification, immunotherapy and photodynamic therapy. In this paper, we first overview the recent applications of Pt-based drugs in antitumor inorganic chemistry, with a focus on summarizing the application of Pt-based drugs and their precursors in the anticancer immune response. The paper also provides a reasonable outlook on the future development of Pt-based drugs from the chemical and immunological perspectives, relying on the existing content and problems of Pt-based drug development. On the basis of the gathered information, joint multidisciplinary programs on implementing comprehensive immune analyses for the future development of novel anticancer metal compounds should be initiated.

**Keywords:** platinum-based antitumor drugs; anticancer immune response; metal drugs; tumor microenvironment regulation; photodynamic and photosensitization therapy

## 1. Introduction

Since Rosenberg's discovery of cisplatin in 1965, cisplatin has quickly reached the peak of its clinical use on account of its potent antitumor properties [1,2]. Generally speaking, Pt(IV) would be reduced into Pt(II) via the reductive microenvironment of tumors. Compared to Pt(IV)-based antitumor drugs, Pt(II)-based ones perform better cytotoxic ability. Cisplatin, for example, has been used extensively in therapies for ovarian, prostate, testicular, lung, nasopharyngeal, esophageal and other cancers through the direct binding of DNA within tumor cells. However, this kind of DNA binding is non-targeted, causing damage to other "healthy" organs and resulting in limitations of its clinical use. Cisplatin has a straightforward structure and a clear mechanism, and it can have some therapeutic effects on many tumor types. However, the substantial side effects (nephrotoxicity, ototoxicity, etc.) and drug resistance brought on by platinum (Pt)-based therapy have progressively come to light, and as the clinical application continues to advance, they have also grown to be a significant barrier to its advancement [3–5]. As a result, second- and third-generation Pt-based medications with the active ingredients carboplatin and oxaliplatin have appeared. Compared to cisplatin, oxaliplatin has moderate adverse effects and could be used for

patients with hepatic dysfunction. It is commonly used in the treatment of colorectal cancer. Third-generation and second-generation Pt-based medications can treat resistance brought by previous-generation Pt-based therapies and have superior stability and fewer side effects compared to cisplatin [6–8]. In summary, the overall development of Pt-based medicines is still constrained by a unique combination of side effects, and the limitations cannot be ignored. [4].

Pt medicines primarily attach to DNA and damage it to have anticancer effects in vivo. For instance, the chloride ion in the structure of the drug cisplatin is activated by hydrolysis as soon as it enters the cell, resulting in the creation of an electrophilic molecule that may covalently attach to the nitrogen atom in the purine residue to damage DNA. Chloride anions improve the reduction potential and ability to receive electrons from some ligands such as DAD [9]. However, the existence of chloride anions could probably improve the cytotoxicity of Pt-based drugs [10]. Pt medications can all stimulate the p53 signaling pathway, caspases and cellular autophagy [11,12]. One of the main factors restricting the therapeutic usage of cisplatin has been its side effects. As a result, in the following years, attention was diverted from the development of Pt(II) to that of its prodrug, Pt(IV) [6]. It had been thought that Pt(IV), which has a gentler and less poisonous structural profile than Pt(II), would serve as the foundation for the creation of new Pt-based medications that might successfully replace Pt(II). Theoretically, reducing biomolecules including glutathione (GSH) and ascorbic acid (ASA) are thought to participate in intracellular reduction processes of Pt(IV) prodrugs to liberate the Pt(IV) parent drug and exert anticancer activity. The slower rate of DNA binding to Pt(IV) compared to Pt(II) is the main reason for the lower in vitro toxicity of Pt(IV) [6,13,14]. The reduction efficiency of Pt(IV) is considered to be a key step in the activation of Pt(IV) complexes during its antitumor activity in vivo.

The development of photoactive Pt-based drugs, immunotherapies and nanomaterials, among other things, as existing Pt-based medications which can reduce with time has received extensive interest [15–17]. In order to lessen side effects, the continual refinement of functional Pt-based drugs is primarily driven by **1**. Reducing adverse responses and improving the selectivity of Pt-based medications **2** can boost effectiveness and synergism and mobilize immunity.

This article focuses on the advancement of Pt-based medications in recent years, listing how these medications have been used to treat malignancies through structural modifications in chemotherapy, photochemotherapy and other treatments. Due to the discovery of the actions of Pt medications on immune cells, including macrophages, their impacts and applications in tumor immunological processes are studied. Nanomaterials, metals, agonists and antagonists are employed in combination with other fields in order to lessen their own hazardous side effects and increase antitumor effectiveness. At the same time, the current advancement of pharmaceuticals based on Pt is coupled with other fields, such as drug design, to offer fresh viewpoints on emerging patterns. Overall, this paper summarizes the development of various Pt-based drugs in recent years, in order to lay some good prerequisites for the research and development of novel Pt-based drugs (Figure 1).

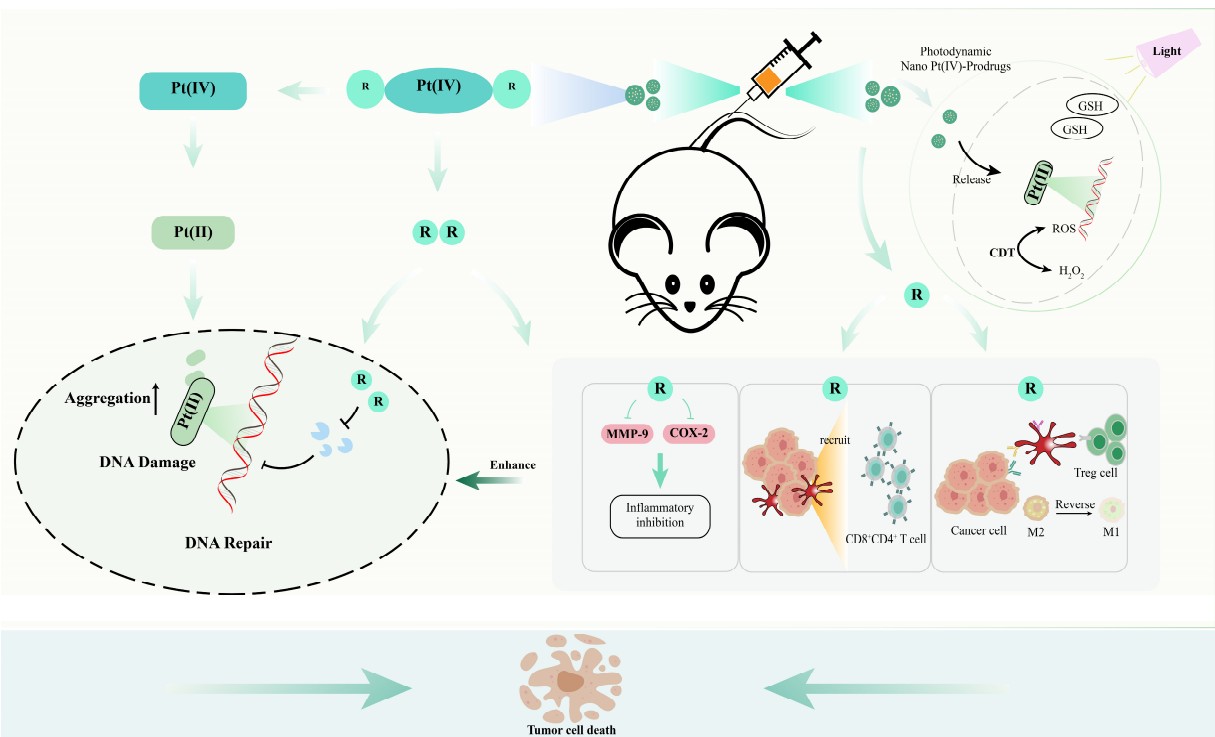

**Figure 1. Multiple pathways to increase the antitumor effect of Pt-based drugs.** The breakdown of nanomaterials or Pt prodrugs to yield Pt(II) drugs, which damage DNA, increase drug aggregation in cells or reduce DNA repair activity, can enhance tumor inhibition or reduce drug resistance. PTT/PDT can facilitate the targeted release of Pt-based drugs to enhance antitumor efficiency or reduce toxic side effects. In addition to their adjuvant role, axial ligands of Pt drugs can be involved in the immune response in vivo. They enhance the antitumor effects of Pt-based drugs by reversing macrophage polarization, recruiting immune cells and inhibiting tumor-associated inflammatory responses.

## 2. Multiple Pathways to Enhance Targeting Properties of Pt-Based Antitumor Drugs

### 2.1. Improving Drug Targeting by Structural Modification of Pt(II) and Pt(IV) Prodrugs

In order to make Pt-based drugs more selective in their action on tumor cells, and at the same time reduce the impact of the adverse effects brought about by the treatment process, many researchers have started to target Pt(II) and Pt(IV) structures by linking them to different groups.

One of the primary methods for creating antitumor medications is to structurally alter Pt-based medicines to match the properties of tumor cells in order to achieve antitumor goals. Solid tumors are characterized by severe hypoxia that is primarily brought on by the hormone hypoxia-inducible factor-1 (HIF-1). Zichen Xu's team [18] used HIF-1 as their target and created a variety of structures that target Pt(IV) alterations. They discovered that YCC-2 (**1**) (Figure 2) greatly reduced HIF-1 expression and improved the effects of cisplatin on HCT-116. Similarly, asparagine synthase plays an important role in tumor growth and metastasis, and as a molecular target, Di Hu's group [19] designed complex **2**, [(bis-NHC)Pt(bt)]PF6 1α (Figure 2), which reduces cellular asparagine levels and effectively inhibits tumor cell proliferation, and a complex that modifies the resistance of cancer cells to cisplatin. Furthermore, heat shock protein 70+ can serve as a recognition site for targeted therapeutics and is closely linked to tumor aggressiveness and therapeutic resistance [20]. Couples **1–5** (Figure 2), designed by A.M. McKeon's group, can target membrane-bound heat shock protein 70+ in cancer cells to increase cytotoxicity [21].

**Figure 2. Improving drug targeting properties by structural modification of Pt(II) and Pt(IV) prodrugs.** (**A**) Structure of cisplatin. (**B**) Structure of oxaliplatin. (**1**) YCC-2. (**2**) Complex [(bis-NHC)Pt(bt)]PF6 1α (**3**, **4**, **5**) compounds designed by A.M. Mckeon's group to target membrane surface heat shock protein 70+. (**6**, **7**, **8**, **9**) Pt(IV) complexes designed by Gao's group with biotin as the axial group. (**10**) Biotin-modified Pt(IV) complex designed by Chen's group. (**11**) DPB structure. (**12**) Dual-targeted Pt(IV) complexes designed by Liu's group.

The addition of biotin to the structure can lead to more precise targeting of Pt-based drugs to cancer cells after structural modifications to increase accumulation in the cells. In 2020, researchers developed four Pt(IV) complexes, **6**, **7**, **8** and **9** (Figure 2), with biotin as the axial group, which were found to have a multiplicative increase in cytotoxicity compared to cisplatin in vitro, as well as a reversal of cisplatin resistance [22]. Xing Wang's group [23] also synthesized Pt(II) complex **10** (Figure 2) to target Pt-based drugs with biotin modifications, which enhanced the antitumor activity. In addition to the increased targeting of biotin, Suxing Jin's group [24] designed and synthesized a new Pt(IV) complex **11**, DPB (Figure 2), which introduced dichloroacetic acid to enhance lipophilicity and cytotoxicity and at the same time impeded the growth of cancer cells with active glycolysis, demonstrating the importance of dual targeting for anticancer effects. The importance of dual targeting for anticancer effects was strongly demonstrated.

Not coincidentally, Xiaomeng Liu's group [25] also relied on dual targeting to design a Pt(IV) prodrug **12** (Figure 2), which was found to enhance intracellular aggregation, thus significantly inducing DNA damage, inhibiting tumor cell migration and effectively suppressing the nephrotoxicity associated with Pt-based drugs. This also improves the strategy for the treatment of advanced postmenopausal breast cancer.

*2.2. Nanoparticulate Pt-Based drug Delivery System to Upgrade Intracellular Accumulation*

Although great progress in nanoparticle-based drug deliveries has been achieved in the past few decades, the toxicity and limitations should not be ignored. In nanoparticle-based drug carriers, liposomes are characterized by self-assembling, and drugs are assembled to be liposome nanoparticles (LNPs). However, liposomes tend to accumulate in the liver, spleen and bone marrow, as well as the mononuclear phagocytic system (MPS) in the human body. Liposomes accumulated in MPS could probably result in serious side effects and toxicity. To address these limitations, improving targeting ability and eliminating influences on other organisms would be the key. Louzhen Fan et al. [26] summarized the perspectives of nanomaterials such as liposomes, proteins and carbon quantum dots as carriers and illustrated the progress of nanoparticle-based drugs in cancer medications.

Based on the above, mesoporous silica nanoparticles incorporating Pt(IV) predrugs were prepared by Zigui Wang et al. [27]. Taking advantage of the liver-targeting properties of lactobionic acid (LA), the nanocarrier enhanced the circulation time while increasing the aggregation effect of the drug in hepatic tumor cells, and as it was in a reducing environment, the bound Pt(IV) could be rapidly reduced for its effect. Li Li's group [28] combined an oxaliplatin prodrug with polyethylene glycol-modified nanosomes and found an enhanced tumor-targeting effect. Because of specific binding to epidermal growth factor receptor (EGFR) nanosomes, accumulation was found to be more pronounced in tumors than in normal cells. In addition, hydrophilic materials and peptide fragments and fluorescent dyes were used as nanocarriers for carboplatin (CRGD), and the complexes were found to be more cytotoxic than carboplatin and to have an appreciable targeting uptake capacity [29].

The combination of targeted therapy and chemotherapy is the main tool used to increase the induction of tumor cell death. It has been clinically found that panitumumab and Pt-based drugs have difficulty accumulating into high concentrations at the tumor site both alone and in combination, which is the main reason for the lack of efficacy. The investigators found that NanoPt-PAN (Figure 3), a nanomedicine made by combining the two agents, was able to improve this phenomenon and was more actively targeted than the single agent [30]. In addition, the authors found that NanoPt-PAN also had good anti-CRC effects, making it a candidate nanomedicine for the treatment of colorectal cancer. Also making nanopolymers out of existing combination regimens is Jianfeng Guo's group [31]: as FNA, 5-FU and oxaliplatin each have low efficacy, high toxicity and long treatment cycles, nanoprecipitation technology was used to design NanoFOLOX, which was found to improve the adverse effects of the three drugs, facilitating blood circulation and the aggregation of Pt-based drugs in tumor cells. The new nanostructures were found to improve the adverse effects of the three drugs and enhance blood circulation and the aggregation of Pt-based drugs within the tumor cells. Furthermore, the authors combined the nanoparticles with a variety of other drugs and found that FOLOX nanoparticles have great potential as the basis for combination therapeutic strategies in the treatment of CRC. Nanoparticles are ideal vehicles for multiple combination strategies due to their multiple drug-carrying capabilities. A novel nanogel system was made by combining CRGDFK-modified nanogels with the sodium channel inhibitors lidocaine and cisplatin, and it was found that the adverse effects produced by the system were alleviated by the targeted release of cisplatin. Due to the introduction of high-affinity peptide fragments, the nanogel showed a significant increase in enrichment at the tumor site, which was more beneficial to the inhibition of primary tumor growth [32].

In the case of brain tumors, such as glioma, the fundamental limitation is the presence of a blood–brain barrier in the brain, which prevents normal drug entry into brain tumor cells and inhibits therapeutic efficacy [33]. Tao Sun's group [34] has developed a novel nanocarrier that has been found to have enhanced targeting and the ability to cross the blood–brain barrier both in vitro and in vivo. The combined studies suggest that this therapeutic strategy could be a new dosage form for Pt-based drugs, particularly for the treatment of clinical gliomas.

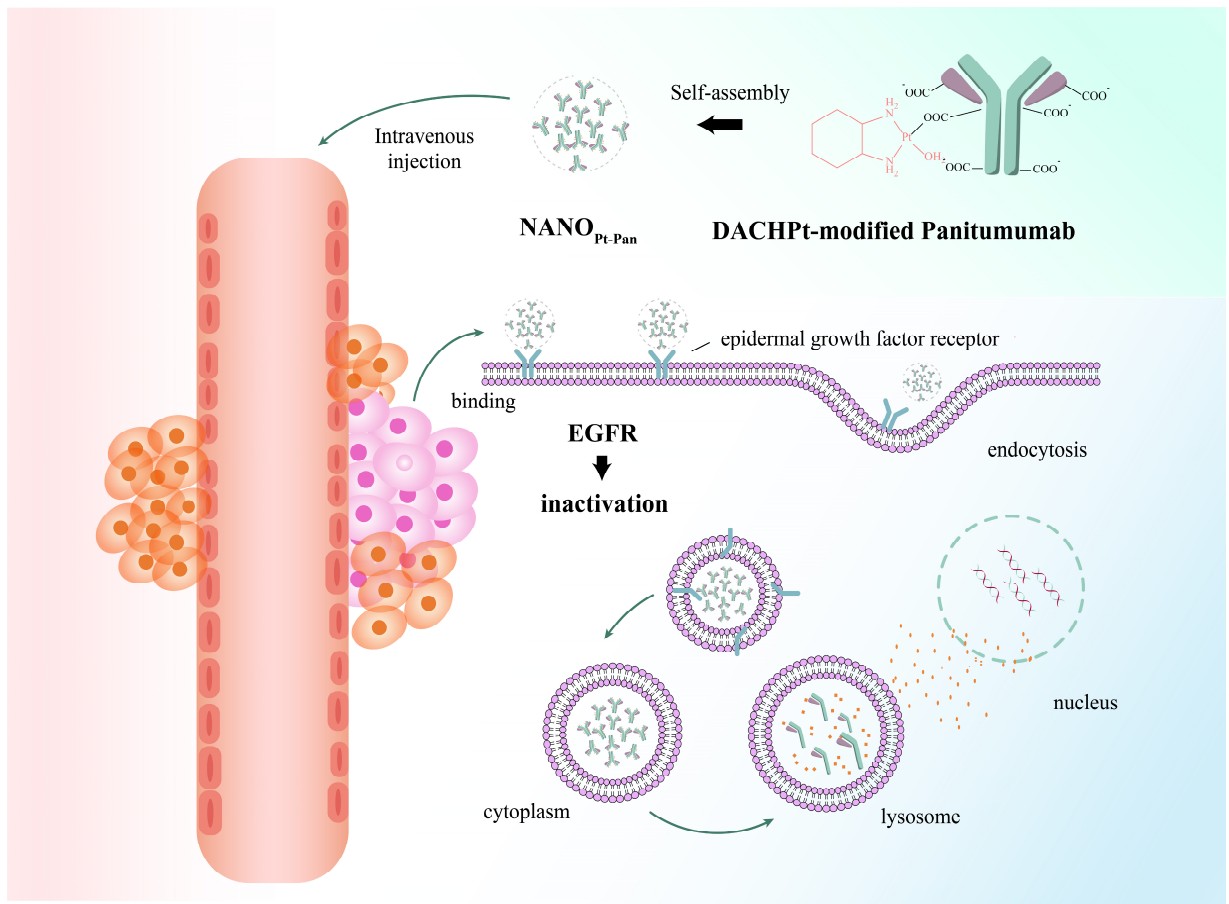

**Figure 3. Scheme of nanoparticulate Pt-based drug delivery system to upgrade intracellular accumulation.** Nanoparticles prevent drug degradation and reduce toxic side effects by delivering the right dose of the drug at the right time and to the right target. Due to the specificity of nanoparticles, researchers are focusing on combining them with Pt-based drugs in order to achieve targeted delivery and increase the accumulation of drugs in tumor cells to achieve the desired efficacy.

In previous studies, it was shown that nanocarriers with tumor acidic activation sites and conversion capabilities have important potential in targeted drug delivery, exhibiting neutral or negative charges in circulation to prolong circulation and promoting cellular internalization for targeted drug delivery. Due to the slow charge reversal at the surface of tumor tissue, Liu et al. [35] prepared UCC-NP/Pt nanocarriers, which can undergo rapid charge conversion at tumor acidity to achieve targeted drug delivery and significant anticancer effects in cisplatin-resistant cells.

In general, the structural modification of existing Pt-based drugs and the use of nanomaterials as drug carriers are promising. The goal is to improve the targeting of the drug and increase the degree of aggregation in tumor cells, thereby providing better antitumor effects or reducing the adverse effects and resistance of Pt-based drugs in clinical use.

## 3. Synergistic Involvement of Pt-Based Drugs in Immunotherapy

The immune system has a role in recognizing and killing tumor cells in antitumor therapy (Figure 4) [36]. Similarly, tumor cells can inhibit the action of immune cells, for example by releasing immunosuppressive factors, or avoid the effect of immune cells by avoiding or weakening their recognition [37]. With the rise of therapeutic approaches targeting immune checkpoints [38], the previous belief that resistance to chemotherapy drugs was due to suppression or disruption of immune system function needs to be re-evaluated. In response to the problems that have arisen in the clinical management of conventional Pt-based drugs, it is hoped that a new range of antitumor Pt-based drugs can be developed and designed to target the immune system. These Pt-based drugs could inhibit the immune escape that occurs during conventional drug therapy, slowing tumor growth and reducing the development of drug resistance. The combination of chemotherapy and immunotherapy will be a promising strategy in the design of this class of drugs.

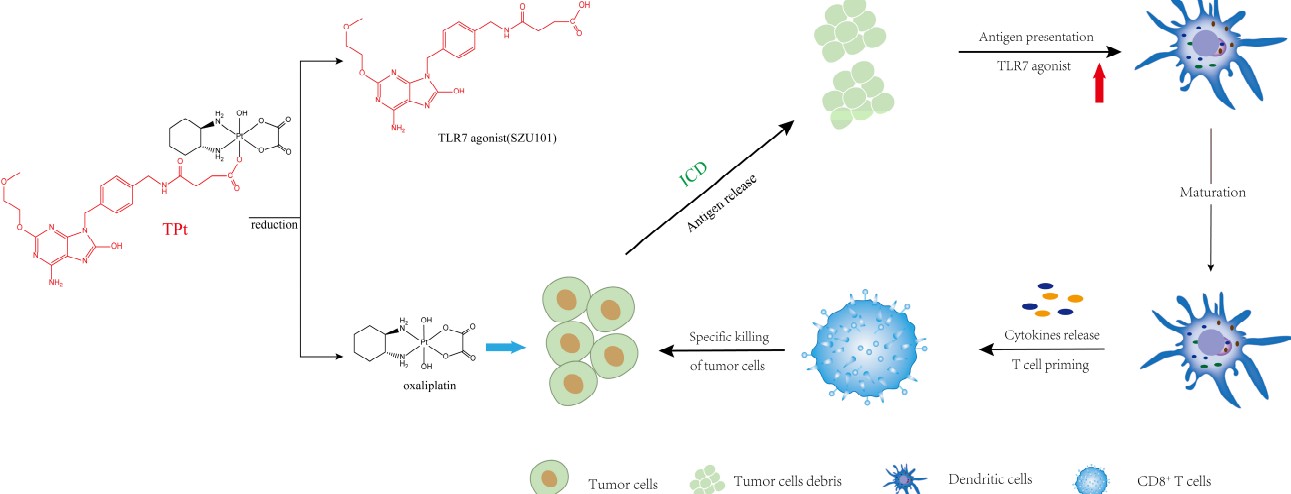

**Figure 4. Scheme of combinations with immune agonists to improve T-cell responses.** Immune agonists, such as a TLR7 agonist, conjugated with oxaliplatin, can accelerate T-cell maturation, activating prime T cells into mature T cells. Mature T cells release cytokines and exert specific killing effects on tumor cells. Antigens, released by tumor cells going through immunogenic cell death, will perform a synergistic effect with immune agonists to improve T-cell responses to tumor cells.

### 3.1. Recruiting Immune Cells to Enhance Immunotherapy Effects

Tryptophan 2,3-dioxygenase (TDO) is an immunosuppressive enzyme that may be involved in the immune escape of tumor cells and the potential for tolerance in vivo [39,40]. Therefore, a Pt(IV) antitumor drug containing a TDO inhibitor was designed to reverse tumor immunosuppression [41]. Flow cytometry suggested that this complex **13** (Figure 5) could induce cell inactivation via the mitochondria-dependent apoptotic pathway and inhibit the TDO enzyme to block the kynurenine pathway downstream, thereby enhancing the immune response of T cells.

Analysis by Takahiro Yamazaki et al. [42] for complex **14**, PT-112 (Figure 5), a novel Pt-based coupling, revealed that the cytotoxic response to PT-112 was associated with exposure of calreticulin on the surface of dying cells, secretion of ATP and HMGB1 and other danger signals that also promoted anticancer immunity. In addition, the authors also found that when combined with an immune checkpoint inhibitor, PT-112 controlled cancer in mice through the systemic immune function of recruiting immunoreactive cells in TME while exerting cytotoxicity, limiting the growth of distant lesions.

Milos's team [43] developed four newly designed Pt-based complexes, **15**, **16**, **17** and **18** (Figure 5), and compared their antitumor effects with cisplatin in vivo and in vitro. The results suggest that [PtCl$_4$(en)] (en = ethylenediamine) increased the number of CD45$^+$ cells

and led to a reduction in metastatic lung lesions in tumor-bearing mice, also highlighting the potential of this structure in antitumor action.

**Figure 5. Synergistic involvement of Pt-based drugs in immunotherapy.** (**13**) Pt(IV) complexes containing TDO inhibitors. (**14**) PT-112. (**15**, **16**, **17**, **18**) Four Pt complexes designed by Milos's group. (**19**) The structure of OPA. (**20**) OXA-NO-enhanced immunotherapy designed by Liu's group. (**21**) ROS-response micelles PKS. (**22**) Novel immunochemotherapeutic agents for the combination of TLR7 agonists with oxaliplatin. (**23**) Naproxen Pt(IV) complexes designed by Han's group. (**24**, **25**) DNP and NP structures. (**26**) Ketoprofen Pt(IV) complex. (**27**) IA-1 structure. (**28**, **29**, **30**, **31**) A family of coumarin derivatives designed by Wang et al.

### 3.2. A Promising Combination of ROS and Macrophages to Direct Polarization Strategy

Tumor-associated macrophages (TAMs) account for the largest proportion of the tumor immune microenvironment (TME) and play a key role in tumorigenesis and progression [44]. Evidence suggests that tumor-associated macrophages accumulate in tumors and lead to drug resistance, whereas cisplatin inhibits the clearance of EGF by tumor-associated macrophages during antitumor therapy, thereby inhibiting tumor progression or recurrence [45]. Andrulis's group [46] studied the treatment of the cisplatin analog "poly-plat", SSP and SAP and found that macrophages treated with "poly-plat" and SSP showed cytoplasmic elongation after stimulation.

M1 macrophages exert their antitumor function through directly mediated cytotoxicity and antibody-dependent cell-mediated cytotoxicity (ADCC) [47]. Cisplatin-loaded umbilical cord-derived exosomes were designed by Xiaohui Zhang's group and found to be several times more toxic and cell-sensitive in drug-resistant cells when compared to chemotherapy alone [48]. Tao Yang et al. [49] found that the Pt-based complex OPA **19** (Figure 5) exerts chemoimmunotherapeutic effects on tumors mainly by blocking DNA replication and inhibiting trigger receptors expressed on myeloid cells 2 (TREM2), while also promoting the polarization of macrophages from M2 to M1. It also stimulates dendritic cells, cytotoxic T cells and natural killer cells to act on cancer cells. In addition, OPA alters the tumor microenvironment to reverse resistance to Pt-based drugs.

Similarly, Zhuang Liu's group designed an epigenetic Pt(IV) complex **20** (Figure 5) to enhance cancer chemoimmunotherapy [50]. This structure also promotes the polarization of macrophages from the M2 to M1 phenotype, thereby reversing the immune microenvironment and reducing immunosuppression, demonstrating better tolerance and inhibition of tumor growth compared to traditional Pt-based drugs. Of course, the investigators found that it is not feasible to rely on immunotherapy alone to inhibit tumor progression and that it needs to be combined with other therapies to maximize the effect of immunotherapy. Therefore, Chun-Liang Lo's group [51] designed an ROS-responsive micelle **21** (Figure 5) based on the restricted regulation of ROS levels in tumor tissue and the importance of ROS in the tumor microenvironment and in the cancer treatment process. It can release Pt-based drugs into the cytoplasm of macrophages and cancer cells, increasing the level of ROS in the tumor and inducing the polarization process towards M1-type macrophages and phagocytosis of cancer cells by macrophages, achieving the dual effect of complementary chemotherapy and immunotherapy.

### 3.3. Combination with Immune Agonists to Improve T-Cell Responses

Myeloid-derived suppressor cells (MDSCs) play a major coordinating role in cancer-associated inflammation, dynamically promoting a differentially polarized inflammatory program in tumor progression that facilitates tumor development and resistance to therapy [52]. As mentioned above, Tao Yang's group found that OPA could alter the tumor microenvironment to reverse drug resistance while inhibiting the expression of TREM2, thus resulting in reduced antitumor responses in mice, including less immunosuppressive macrophages, more secretion of immunostimulatory molecules and improved T-cell responses [49].

The synergistic effect of Pt-based drugs and immunotherapy was even more effective in drug-resistant cancer models, and Zhigang Wang's group [53] designed a novel immunotherapeutic agent **22** (Figure 5) by binding TLR7 agonists to oxaliplatin prodrugs, which induced immunogenic death of 4T1 cells and activated dendritic cells to secrete proinflammatory factors such as IFN-γ, TNF-α, IL-6 and IL-12. The mechanism suggests that intratumor cytotoxic T cells are activated, thereby increasing antitumor efficiency.

### 3.4. Pt-Based Drugs Combined with NSAIDs to Repress Tumor-Related Inflammation

Inflammation is an important feature of cancer progression and is associated with the development, progression and metastasis of malignancies; therefore, the inhibition of the associated inflammatory response plays an important role in antineoplastic chemother-

apy [54,55]. The introduction of NSAIDs into Pt-based regimens has been shown to significantly improve the efficacy of NSAIDs compared to cancer treatment alone [56–58]. Han's group [59] used naproxen in combination with Pt(IV) and found that compound **23** (Figure 5) had better antitumor properties, with a Pt-based fragment in its structure causing DNA damage and naproxen as a non-steroidal anti-inflammatory agent inhibiting COX-2 and reducing the associated inflammatory response, while both significantly inhibited MMP-9 activity in tumors, thereby helping to reduce the growth and metastasis of aggressive tumors. Therefore, in line with the trend towards the development of nanodrug delivery systems in oncology therapy, Linming Li et al. [60] developed new bovine serum protein nanoparticles based on naproxen Pt-based complexes, which were found to have better antitumor effects and lower toxic side effects than the free compounds. In addition, the addition of nanoparticles reduced tumor inflammation targeting COX-2, MMP-9 and iNOS, which was more beneficial to the antitumor capacity and provided a new clinical approach to overcome the shortcomings of Pt-based drugs. In addition, Wang's group [61] prepared two Pt-based complexes, DNP **24** (Figure 5) and NP **25** (Figure 5), with naproxen as the axial phase ligand and found that DNP showed potent antitumor activity and reduced toxicity in triple-negative breast cancer mice. DNP also reduced prostaglandin secretion and inhibited c-Myc expression, showing that the complexes could interfere with the inflammatory and metastatic processes of breast cancer.

As another NSAID approach, Zuojie Li and Qingpeng Wang et al. [62] designed and prepared a ketoprofen Pt(IV) complex **26** (Figure 5) which, in addition to the several effects described previously, notably inhibited PD-L1 expression, improving immune responses and CD8[+] T-cell infiltration in tumor tissues. In 2022, Gou's group [56] designed and found, through etodolac binding to Pt(II), that LA-1 **27** (Figure 5) was significantly cytotoxic and inhibited metastasis of A2780 cells by inhibiting the COX-2/JAK2/STAT3 axis.

Since the structure of coumarin confers anti-inflammatory effects on its derivatives by inhibiting COX activity, Bingquan Wang et al. [63] designed a series of Pt-based coumarin derivatives, **28**, **29**, **30** and **31** (Figure 5), and found that in addition to their respective structural effects, they could also induce apoptosis by upregulating the expression of caspase 3 and caspase 9. The group also evaluated the anticancer activity of bifunctional 7-hydroxycoumarin Pt(IV) and found that the derivatives could release Pt(II) compounds to attack DNA and inhibit COX activity, which has great potential to overcome Pt(II) resistance [64].

## 4. Multi-Targeting Structure Modification to Actualize Diverse Antitumor Actions

Tumor tissues are often generated by kinds of mutations, and there are significant differences between samples of the same type of tumor from the same patient, so scholars have tried to achieve more satisfactory results by combining therapies that act on multiple cancer targets or by structurally modifying a drug [65]. For example, clinical attempts to use "cocktail therapy" for the treatment of AIDS beginning as early as the last century and the combination of chemotherapy and immunotherapy to improve antitumor outcomes as described above have the same aim of using multiple drugs with different effects in a more comprehensive and synergistic manner to achieve optimal efficacy for a particular condition. This section focuses more on the recent work on structural modifications of Pt-based drugs to achieve dual- or multi-targeting effects and provides more data to support the enhancement of the antitumor effects of Pt-based drugs.

Kogularamanan Suntharalingam et al. [66] reported a planar Pt(II) complex [Pt(BDI(QQ))]Cl, which the authors found to have a dual targeting ability, acting on both DNA and mitochondria. [Pt(BDI(QQ))]Cl induces DNA damage and acts selectively on cancer cells; it can also accumulate in mitochondria to cause direct damage. The article also suggests that p53 is not a determinant of complex activity and therefore can be targeted to cancers with a p53-deficient state. In the same year, the group [67] also synthesized two Pt(IV) complexes, **32** and **33** (Figure 6), of vitamin E and a-TOS and found that OET has a dual targeting effect in killing cancer cells, with the Pt-based group causing DNA

damage and the axial ligand causing mitochondrial dysfunction, further validating the value of a dual targeting strategy. Weike Su's group [68] designed and synthesized four novel Pt(IV) complexes and found that compound **34** (Figure 6) could release Pt(II) and DCA within the tumor cells, leading to DNA damage as well as disruption of mitochondrial membrane potential, illustrating the great potential of **34** in dual-targeted antitumor action. In addition, Mariafrancesca Hyeraci et al. [69] have prepared Pt(II) complexes **32** and **33** with a triphenylphosphorus fraction based on previous studies. This makes the trans-[PTBR2(NHRR')(PPH$_3$)] proposed in this article an option for the design of multi-targeted oncology drugs.

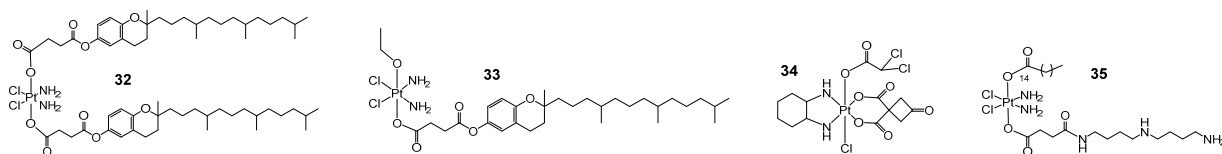

**Figure 6. Multi-targeting structure modification to actualize diverse antitumor actions. (32, 33)** Pt(IV) complexes of vitamin E and a-TOS. **(34)** Structure of compound designed by Su's group. **(35)** Polyamine-Pt(IV) prodrug.

In addition to targeting DNA damage and mitochondrial damage in tumor cells, Jin-Lei Tian et al. [70] designed a series of dual-targeted monofunctional Pt(II) complexes based on the aspect of improving drug delivery and targeting during antitumor therapy. This series of compounds can be used as one of the targets to rapidly enter the cell through the upregulation of Gluts in tumor cells leading to increased glucose uptake, and then to induce apoptosis through lysosomal damage caused by the production of large amounts of reactive oxygen species (ROS) through the localization of the P-gp protein.

Polyamines play an important role in tumor growth, progression and migration and have also been found to be highly expressed in tumor cells. Targeting polyamines to block their metabolism has become one of the popular anticancer drug targets [71]. Summarizing the previous studies, Liu's group [72] has designed polyamine-Pt(IV) prodrug **35** (Figure 6), which can effectively inhibit tumor growth and reverse resistance to cisplatin. In addition to targeting DNA, complex **35** can also alter the high-polyamine environment and inhibit tumor cell growth by upregulating SSAT and PAO and downregulating putrescine, spermine and spermidine concentrations.

## 5. Activation of Pt-Based Prodrugs via Thermal/Invisible Light Stimuli

Photothermal and photodynamic therapies have become a popular area of interest in oncology treatment in recent years. Researchers have experimented with different structures of photosensitive materials and nanomaterials and have demonstrated in a variety of cancer cells and tissues that PDT and PTT are among the factors to be considered in the antitumor process due to their uniquely targeted low toxicity and high stability.

### 5.1. NIR-Based Photothermal Therapy Using Pt-Based Drugs

Near-infrared fluorescence imaging (Figure 7) [73] has features such as deep penetration and low photodamage among current imaging techniques [74]. Therefore, combining it with Pt-based drugs is a preferred choice for the preparation of novel photothermal therapy drugs. Near-infrared nanodrug encapsulation is a new technique for reversing tumor resistance to Pt-based drugs. Chengwei Zhang et al. [75] investigated the use of melanin-like nanoparticles (MENPs) in combination with Pt(IV) to obtain targeted therapy for prostate cancer. The authors found that the nanomaterials had appreciable biocompatibility, antitumor activity and conversion efficiency. The synergistic effect of the two resulted in a significant improvement in the antitumor effect. Zhang's group [76] proposed a way to overcome cisplatin resistance through NIR photothermal therapy by means of a prepared nanosystem F-Pt-NPS **36**, **37** (Figure 8). The authors found that NIR laser-induced

nanocomplexes could promote drug uptake while accelerating GSH depletion to activate cisplatin; they also found that these nanocomplexes could increase cisplatin's cross-linking to DNA and inhibit DNA repair. In addition, they showed good tumor inhibition in vitro and in vivo and the reversal of cisplatin resistance, suggesting an important idea of using infrared light-induced mild heat therapy to address cascade resistance.

Furthermore, in order to improve drug targeting effect, Yiyun Cheng et al. [77] reported a nanoparticle that has the anticancer activity of phytic acid (PA) while maintaining the photothermal effect of Pt-based nanoparticles by using natural PA-modified Pt-based nanoparticles with bone targeting properties combined with hydroxyapatite. Under NIR light irradiation, the growth of bone tumors and tumor-associated osteolysis were effectively inhibited. Also to increase targeting, Lianshuai Gu et al. [78] synthesized folic acid-modified cisplatin-loaded ICG lipid–polymer hybrid nanoparticles, FCINPS, known to be FDA-approved near-infrared fluorescent dyes. Induced apoptosis and necrosis also provide good support for tumor-targeted therapeutic nanopharmaceutical formulations.

Mao's group [79] has developed a biotin-labeled Pt(IV) prodrug **38**, **39** (Figure 8) to address cisplatin resistance in tumor cells through mitochondrial targeting and photothermal therapy. Pt(IV)-NPs were able to enhance mitochondrial targeting to induce maximal mitochondrial DNA damage and increase the intracellular accumulation of Pt, while also reducing GSH levels and inhibiting DNA repair, with significant inhibitory effects on A549cisR cells in the presence of targeted chemotherapeutic–photothermal synergistic treatment.

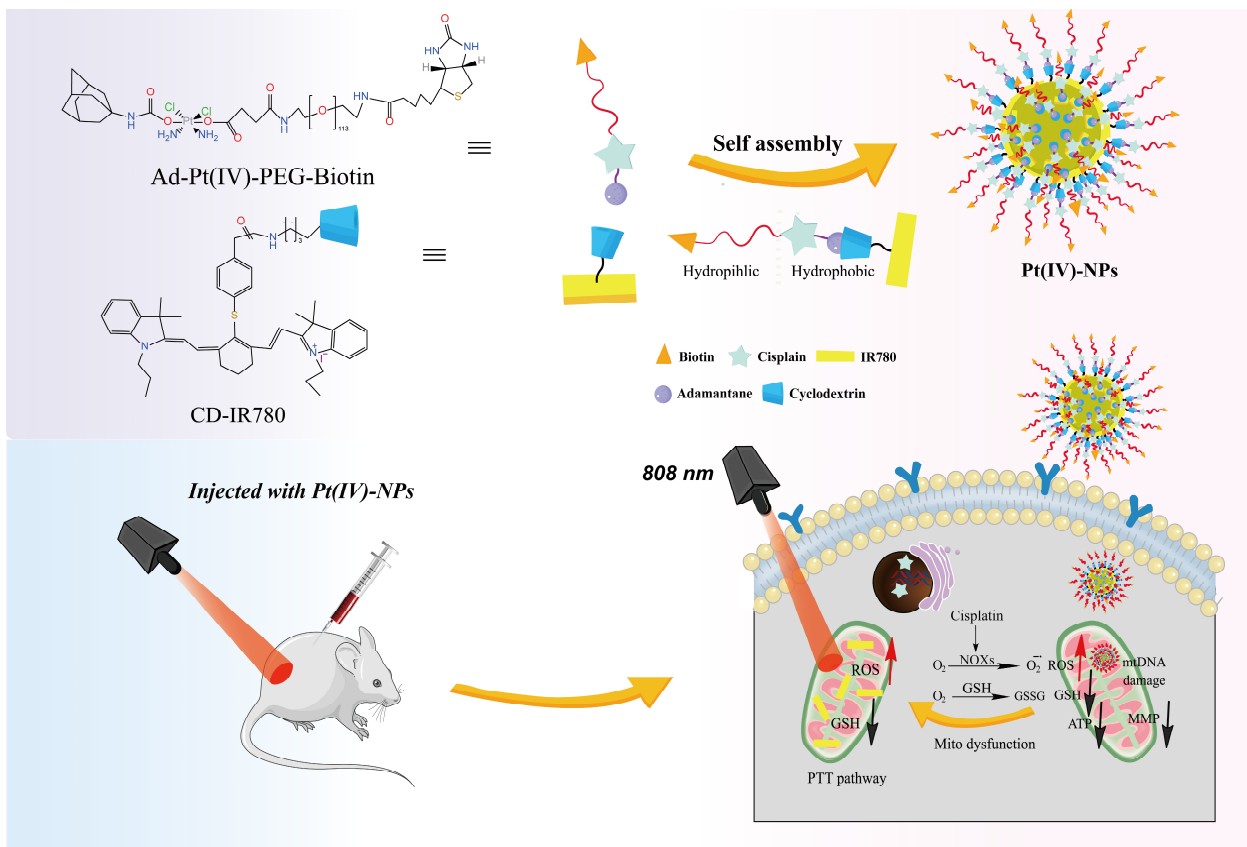

**Figure 7. Mechanism of NIR-based photothermal therapy in Pt-based drugs.** This structure has both hydrophilic and hydrophobic ends, which is the structural basis module for this hybrid drug's self-assembling into nanoparticles, making the intracellular accumulation easier. Via NIR Irradiation, this basic module will be degraded into 5 parts, which then target the nucleus and mitochondrion, resulting in ROS enhancement and GSH reduction. In addition, there also exists mt DNA damage and MMP decline in the mitochondrion.

**Figure 8. Structures of Pt-based prodrugs activated via thermal/invisible light stimuli.** (**36, 37**) The Zhang group obtained the prepared nanocomplexes by systematically combining P1 (**36**) with DAP-F (**37**) and Pt. (**38, 39**) Pt(IV)-NPs are mainly composed of IR1780 (**38**) and Ad-Pt(IV)-PEG-biotin (**39**). (**40**) The structure of prodrug. (**41, 42, 43**) CPNP-Fc/Pt consists mainly of DSPE-PEG500-NH$_2$ (**41**), Pt(IV) prodrug (**42**) and CP (**43**). (**44, 45, 46**) (CAT)@Pt(IV)-liposome consisting of DSPE-Pt(IV) (**44**), DSPE-PEG5K (**45**) and DPPC (**46**).

## 5.2. Pt-Based Drugs for Oxygen-Dependent Photodynamic Therapy

The hypoxic environment of solid tumors can limit the production of reactive oxygen species (ROS) by photodynamic therapy, thereby limiting its effect. Dongbo Guo et al. were the first to polymerize Pt(IV) complex precursor monomer (PPM) with 2-methacryloyloxyethyl phosphorylcholine (MPC) to form a nanoprecursor **40** (Figure 8) that could be reduced to Pt(II) under light irradiation, demonstrating that high levels of ROS could be produced in the absence of endogenous oxygen. The article also demonstrates that this structure has a long half-life and high aggregation in the antitumor process and can downregulate the expression of multidrug resistance-associated protein 1 (MRP1), thereby reversing the drug resistance problem in tumor cells. Chemodynamic therapy relies on the involvement of transition metal ions and endogenous H$_2$O$_2$, but low levels of endogenous H$_2$O$_2$ can also affect the efficacy of CDT [80]. The Ju group [81] developed CPNP-Fc/Pt **41, 42, 43** (Figure 8), a nanoparticle used to increase local oxygen levels and enhance the effects of CDT. The article illustrates that the main mechanism is that the release of Pt(II) from glutathione simultaneously triggers a cascade reaction of NADPH oxidase and superoxide dismutase to increase H$_2$O$_2$ levels to ensure an effective supply of H$_2$O$_2$, showing good CDT effects and inhibiting tumor growth. Zhuang Liu et al. [82] encapsulated CAT in Pt(IV) phospholipids to make CAT@Pt(IV)-liposome **44, 45, 46** (Figure 8) for enhancing the effect of tumor chemotherapy. The authors found that the use of this liposome resulted in good protection of enzymatic activity and triggered the breakdown of H$_2$O$_2$ in tumor cells to alleviate

hypoxia in the environment, with good synergistic effects, providing a novel option for the clinical treatment of tumors. Similarly, to enhance the $H_2O_2$ generation of ROS to diminish cancer cells, some scholars used endogenous $H_2O_2$ for the conversion of reactive oxygen species to enhance the antitumor effect by using Fenton chemistry [83]. In this paper, an organic nanomedicine PTCG NPs was designed using EGCG, a phenolic Pt(IV) prodrug (Pt-OH) and a polyphenol-modified block copolymer (PEG-b-PPOH) as a backbone to achieve efficient drug release after cellular internalization. After being activated by the release, the cisplatin in the nanomedicine acts as an artificial enzyme involved in a cascade reaction to produce $H_2O_2$ for CDT and catalyzes the generation of highly toxic reactive oxygen species in the Fenton reaction to produce good anticancer effects. In addition, the avoidance of toxic side effects of Pt-based drugs provides a powerful strategy for cascade cancer therapy with nanomedicines.

## 6. Complexes with DNA Expression and Histone Post-Translation Depressants

Conventional Pt-based drugs achieve their antitumor effects primarily by acting on DNA damage. However, tumor cells can also reduce DNA damage or repair DNA through strategies such as translesion synthesis (TLS), nucleotide excision repair (NER), homologous recombination (HR) and other pathways in addition to increasing exocytosis [84,85]. Tumor cells can develop resistance to antitumor drugs through the repair of DNA damage [86]. While the effects of drug-resistant tumors of increasing accumulation in tumor cells and acting on immune pathways to reverse tumor cell immune escape have been described previously, this section focuses on the reduction of tumor cell resistance by blocking the DNA repair process and inhibiting genetic materials.

### 6.1. Histone Acetylation (HDAC) Inhibitors to Reverse Drug Resistance in Tumor Cells

Post-translational modifications of histones are not only involved in dynamic processes such as transcription and DNA repair in cells but are also associated with the maintenance of chromatin stability. Alterations in histones can affect DNA repair, mitosis and meiosis, and the inhibition of histones can be used to target DNA in tumor cells [87,88]. Therefore, histone acetylase inhibitors (HDACi) are novel anticancer drugs that inhibit histone acetylation.

A series of HDACi-containing Pt(IV) prodrugs, **47**, **48**, **49** and **50** (Figure 9), were designed and prepared by Zichen Xu et al. for multiple targeting of genomic DNA, histone acetylase and PARP-1 [89]. The article illustrates the significant antiproliferative activity of the prepared Pt(IV) prodrugs against cisplatin-resistant tumor cells. These compounds mainly resulted in increased DNA damage in chromatin and inhibited the repair process. Dan Gibson's group prepared a series of double- and triple-acting Pt(IV) analogs **51**, **52** and **53** (Figure 9) based on the currently more effective Pt(IV) complex satraplatin, cct-[Pt(NH3)(c-hexylamine)Cl2(OAc)$_2$], in order to overcome satraplatin analog drug resistance, most of which exhibited improved water solubility in the analogs [90,91]. They found that one of the triple-acting compounds, **51** (Figure 9), was active in all cell lines, causing DNA damage to induce apoptosis. In addition, targeting HDAC and nuclear DNA is a promising therapeutic strategy.

**Figure 9. Structures of complexes with DNA expression/histone post-translation related depressants. (47, 48, 49, 50)** Pt(IV) prodrug structure containing HDACi. **(51, 52, 53)** Three satraplatin analogs designed by Dan Gibson's group. **(54, 55)** A prodrug for the simultaneous activation of Pt as well as the redox polymer of camptothecin. **(55, 56)** Structures of DDNPs (**55**) and SNNPs (**56**). **(57, 58)** Two Pt(IV) prodrugs containing PARPi3-aminobenzamide (3-ABA) fragments. **(59, 60)** The supramolecular combination chemotherapy system (DOX@PtC10-CP6A) was prepared from CP6A (**59**) and Pt(IV) to obtain an amphiphilic aggregate (PtC10-CP6A) (**60**) which was then wrapped in DOX. **(61)** Octahedral coordination of Pt(IV) predrugs. **(63, 64)** Components of dual drug co-delivery systems.

### 6.2. DNA Expression Inhibitors of Various Levels to Enlighten Novel Modifications

Based on the shortcomings of existing nanomedicines, combined with the selective cytotoxicity of camptothecin on cells with S-phase DNA, Qixian Chen's group [92] proposed the selective and simultaneous activation of Pt-based and camptothecin redox-reactive polymeric predrugs in the cytoplasm **54**, **55** (Figure 9), and these drugs are self-assembled into a single nanoparticle by appropriate chemical methods. In the cytoplasm, multiple predrugs are activated and released in close proximity to the drug targets to exert an optimal therapeutic effect. The redox effect of the predrugs also depletes endogenous GSH

and promotes greater sensitivity of tumor cells to chemotherapeutic agents. Hongtong Lu and Shasha He et al. [93] designed DDNPs **56** (Figure 9) and SNNPs **57** (Figure 8) as light-activated Pt-based synergistic chemotherapeutic dual-sensitive dual precursor drug nanoparticles, which are photosensitive and can activate Pt(IV) to Pt(II) under UVA light while producing a small amount of $N_3$, which contributes to the lysosomal escape of DNNPs to achieve better photoactivated chemotherapy. In addition, the acid-sensitive protein phosphatase 2A inhibitor desmethyldeoxorubicin (DMC) in an acidic microenvironment can block DNA repair pathways and reverse resistance to Pt-based drugs.

PARP inhibitors, the first clinically approved antitumor drugs that utilize synthetic lethality, primarily target poly ADP-ribose polymerase [94]. This inhibitor can exert antitumor effects by inhibiting reparation of DNA damage and promoting apoptosis in tumor cells, and one of the reasons for resistance to CDDP is the enhanced DNA repair activity of tumor cells [95,96]. Mauro Ravera's group [97] combined the two structures and introduced two Pt(IV) prodrugs **58**, **59** (Figure 9) containing PARPi3-aminobenzamide (3-ABA) fragments and found their therapeutic effect was superior to that of CDDP, and the mechanism of action was found to be a combination of DNA damage by CDDP and inhibition of the repair process by PARPi resulting in a potent antitumor effect.

### 6.3. Conjugations of Doxorubicin (DOX) and Pt-Based Drugs to Reduce Drug Resistance

DOX is an inhibitor of RNA and DNA synthesis in tumor cells. Chen's group prepared amphiphilic aggregates (PtC10-CP6A) using supramolecular carboxylated columnar aromatics (CP6A) and Pt(IV) and encapsulated DOX in them to obtain a supramolecular combination chemotherapy system (DOX@PtC10-CP6A) **60**, **61** (Figure 9). The drug can be selectively released at the specific pH of the TME and was found to be non-toxic to normal cells. However, compared to CDDP and DOX, this drug not only inhibits tumor progression but also reduces the toxic effects of CDDP itself [98–100]. In addition, Tang's group assembled a synergistic drug delivery system **62** (Figure 9) using Pt(IV) with octahedral coordination and DOX, which has a dual blocking effect on nuclear and mitochondrial genetic material, and the nanoshell enhances targeting and in vivo accumulation, thus significantly improving the therapeutic effect compared to CDDP and DOX [101,102]. Furthermore, Caiying Zhu and Jingjing Xiao et al. [103] developed a novel DOX and Pt(IV) dual drug co-delivery system **63**, **64** (Figure 9) using the amphiphilic block copolymer PCL-b-p(OEGMA-co-AzPMA) as a nanoscale drug carrier. The substance showed a 2-5-fold increase in killing effect compared to CDDP and DOX in HeLa cells and A357 cells.

### 7. Conclusions and Outlook

CDDP, oxaliplatin and a series of Pt(II) class metallo-antitumor drugs have many limitations in clinical use, including low selectivity, poor accumulation in vivo, drug resistance and adverse effects caused by long-term use. Pt(IV) drugs have become a hot topic of research in recent years as modifying their axial position can reduce these problems. At present, the main research directions for Pt drugs are structural modifications and the selection of nanomaterials, and considerable progress has been made. By combining them with structures with different properties, scholars have achieved significant results in improving the efficiency of antitumor agents, reversing the resistance of tumor cells to Pt drugs and reducing adverse effects. Photodynamic and photothermal therapies, which are widely used in nanomaterials and other areas, also offer many options to improve the targeting and biostability of Pt-based drugs. Meanwhile, the development of Pt-based drugs is not limited to chemotherapy, as researchers are focusing on the synergistic effects of a combination of immunotherapies in order to achieve multiple targets on tumor cells and tissues and to explore the possibility of a comprehensive inhibition of tumor progression. Although tumorigenesis is a combination of many factors, in general, Pt-based drugs still have a large potential for development in oncology treatment. Future development of Pt-based drugs should focus on synergistic multi-treatment, with immunotherapy providing strong support for antitumor chemotherapy, effective modulation of the tumor immune microen-

vironment, mobilization of specific immune cells, etc., to achieve key tumor cell-targeting effects and reduce the incidence of drug resistance and adverse reactions to Pt-based drugs.

**Author Contributions:** Conceptualization, J.M., S.X., C.W., Y.C. (Yutong Chen), R.M., T.D. and T.L.; methodology, J.M., R.M., S.S., P.Z., Y.C. (Yutong Chen) and R.M.; software, J.L., Y.C. (Yi Cao), B.H., T.L., W.Z. and T.D.; investigation, J.L., Y.C. (Yi Cao), B.H., Z.Z., J.G., H.N., W.Z. and J.W.; writing— original draft preparation, J.L. and Y.C. (Yi Cao); writing—review and editing, J.L., Y.C. (Yi Cao) and B.H.; visualization, J.L., Y.C. (Yi Cao), B.H. and T.L.; supervision, J.M., S.X., C.W. and P.G.W.; project administration, J.M., S.X. and P.G.W.; All authors have read and agreed to the published version of the manuscript.

**Funding:** This work was supported by the China Postdoctoral Science Foundation (Grant 2021M701089), Key Scientific Re-search Projects in Henan Colleges and Universities (Grant No. 22A350002), the key scientific research projects of universities in Henan province (222102310402 and 222102310216), Key Program "New Drug Creation" of Guangdong Key Research and Development Plan (No. 2019B020202001).

**Data Availability Statement:** Not applicable.

**Conflicts of Interest:** The authors declare no competing financial interests.

## Nomenclature

| | |
|---|---|
| GSH | glutathione |
| ASA | ascorbic acid |
| HIF-1 | hypoxia-inducible factor-1 |
| LA | lactobionic acid |
| EGFR | epidermal growth factor receptor |
| CRGD | carboplatin |
| TDO | tryptophan 2,3-dioxygenase |
| TAM | tumor-associated macrophage |
| TME | tumor immune microenvironment |
| ADCC | antibody-dependent cell-mediated cytotoxicity |
| TREM2 | trigger receptors expressed on myeloid cells 2 |
| MDSC | myeloid-derived suppressor cell |
| ROS | reactive oxygen species |
| MENPs | melanin-like nanoparticles |
| PA | phytic acid |
| PPM | precursor monomer |
| LNP | liposome nanoparticle |
| TLS | translesion synthesis |
| NER | nucleotide excision repair |
| HR | homologous recombination |
| HDAC | histone acetylation |
| HDACi | histone acetylase inhibitors |
| PARP-1 | poly ADP-ribose polymerase-1 |
| DOX | doxorubicin |
| CP6A | carboxylated columnar aromatics |
| 3-ABA | 3-aminobenzamide |
| PDT | photodynamic therapy |
| PTT | photothermal therapy |
| CDT | chemodynamic therapy |
| UVA | ultraviolet-A |
| MRP1 | multidrug resistance-associated protein 1 |
| MPC | 2-methacryloyloxyethyl phosphorylcholine |
| MPS | mononuclear phagocytic system |

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
