# Peer review of "Older but Stronger: Development of Platinum-Based Antitumor Agents and Research Advances in Tumor Immunity"

_inorganics, doi:10.3390/inorganics11040145_

Round 1
Reviewer 1 Report
This review manuscript is a very good example of a comprehensive review of a specific field of research. The authors very carefully overview new directions in Pt anticancer complexes: synthesis methods, activity and mode of action study. This review indeed will be interesting to many readers of the journal. However, from the chemistry point, all drawings in the manuscript have some sort of mistakes and this diminishes the good impression of the huge work done by analysing all data. Please check all figures! In almost all figures, you have mixed-style drawings and mistakes in formula. For example, cisplatin core, in some figures with the representation of geometry in some do not. You should have only one style for drawing similar structures. Same for the oxaliplatin core, in some examples even no stereochemistry was provided. Authors must carefully check all figures for the revised version.
Author Response
Comment 1: This review manuscript is a very good example of a comprehensive review of a specific field of research. The authors very carefully overview new directions in Pt anticancer complexes: synthesis methods, activity and mode of action study. This review indeed will be interesting to many readers of the journal.
Answer: Thanks for the reviewer’s positive evaluations.
Comment 2: However, from the chemistry point, all drawings in the manuscript have some sort of mistakes and this diminishes the good impression of the huge work done by analyzing all data.
Answer: Thanks for the reviewer’s kind reminder. With the reviewer’s suggestions, we have revised all the drawings of structures in the manuscript to make a better expression of the huge work.
We transformed the structural formulas of complex 3, 4, 5, 6, 7, 8, 9, 10, 11 and 12 in Figure 2, the structures of complex 14, 19, 22, 23, 24, 25, 30 and 31 in Figure 5, the structure of 34 in Figure 6, the structures of complex 39, 40, 41 and 42 in Figure 8 as well as the structures of complex 55 and 64 in Figure 9 into plantar ones, as is shown in the appendix file.
Comment 3: Please check all figures! In almost all figures, you have mixed-style drawings and mistakes in formula.
Answer: Thanks so much for the reviewer's kind reminder. With the reviewer’s suggestion, we carefully checked all the figures according to the articles we referred to and unified the style of conjugations within the manuscript. We have unified all the structural formulas to be planar ones. Mixed-style drawings and mistakes have already been corrected. Figure 2, for example, has been displayed in a new, more brief, and organized layout, as shown below. The rest of other drawings have been reorganized as we did in Figure 2.
Comment 4: For example, cisplatin core, in some figures with the representation of geometry in some do not. You should have only one style for drawing similar structures. Same for the oxaliplatin core, in some examples even no stereochemistry was provided. Authors must carefully check all figures for the revised version.
Answer: Thanks for the reviewer’s kind reminder. As for the oxaliplatin core, we chose not to provide a stereochemistry structure in order to unify the style of all drawings including the oxaliplatin core, cisplatin core, as well as some similar structures.

Reviewer 2 Report
The review of Songqiang Xie and co-authors is about platinum(II or IV)-based coordination compounds or nano-particles with anti-tumor biological properties. The review has a lot of good quality chem schemes which help to navigate and understand manuscript. The authors made a really good job. The current review is an interesting work and I certainly recommend accepting it for publication in the journal Inorganics. This work will definitely attract a lot of attention from researchers and increase citations for the Inorganics.
I have a few minor questions.
Page 2, section 2.1. “For instance, the chloride ion in the structure of the drug cisplatin is activated by hydrolysis as soon as it enters the cell, resulting in the creation of an electrophilic molecule that may covalently attach to the nitrogen atom in the purine residue to damage DNA.” However, if we look at the modified platinum complexes (Figure 2), we see that not all platinum metallocentres contain chloride anions. How important is it nowadays to keep the platinum ion surrounded by two chloride anions and two nitrogen atoms? Like in these articles: 10.3390/molecules27238565 or 10.1021/acs.inorgchem.1c03314?
I really liked the short section 2.2. about platinum nanoparticles. It is not entirely clear to me whether the examples presented were in vivo or in vitro? Only about one example was a comment about in vivo. If there were in vivo studies, I would very much like to look at the LD50 or LC50 data, and has the biodynamics of platinum nanoparticles been studied in living organisms? Were the nanoparticles carcinogenic by any chance?
Page 11, “In addition to targeting DNA damage and mitochondrial damage in tumour cells, Jin-Lei Tian et al. [67] designed a series of dual-targeted monofunctional Pt(II) complexes 1-8 based on the aspect of improving drug delivery and targeting during antitumour therapy.” Compounds 1-8 came from the original paper – need to another numerate or delete the information.
Author Response
Comment 1: The review of Songqiang Xie and co-authors is about platinum(II or IV)-based coordination compounds or nano-particles with anti-tumor biological properties. The review has a lot of good quality chem schemes which help to navigate and understand manuscript. The authors made a really good job. The current review is an interesting work and I certainly recommend accepting it for publication in the journal Inorganics. This work will definitely attract a lot of attention from researchers and increase citations for the Inorganics.
Answer: Thanks for the reviewer’s positive evaluations.
Comment 2: I have a few minor questions.
Comment 2.1. Page 2, section 2.1. “For instance, the chloride ion in the structure of the drug cisplatin is activated by hydrolysis as soon as it enters the cell, resulting in the creation of an electrophilic molecule that may covalently attach to the nitrogen atom in the purine residue to damage DNA.” However, if we look at the modified platinum complexes (Figure 2), we see that not all platinum metallocentres contain chloride anions.
Answer: We greatly appreciate the careful reading and constructive comments from this reviewer, thus We double-checked the complexes in Figure 2, and reorganized the layout of this Figure. The revised version was shown in the file below.
Comment 2.2. How important is it nowadays to keep the platinum ion surrounded by two chloride anions and two nitrogen atoms?
Like these articles, 10.3390/molecules27238565 or 10.1021/acs.inorgchem.1c03314?
Answer: Thanks for the reviewer’s valuable comments and we really appreciated the two documents the reviewer recommended above. As a result, we reconsidered the importance of two chloride anions and two nitrogen atoms surrounding the platinum core, based on the two articles mentioned by this reviewer.
As described in the article "α-Diimine Cisplatin Derivatives: Synthesis, Structure, Cyclic Voltammetry, and Cytotoxicity", chloride anions improve the reduction potential and ability to receive electrons from some ligands like DAD, etc. However, as is demonstrated in "Heteroleptic Pd(II) and Pt(II) Complexes with Redox-Active Ligands: Synthesis, Structure, and Multimodal Anticancer Mechanism" published in Inorganic Chemistry, the existence of chloride anions could probably improve the cytotoxicity of Pt-based drugs.
Thus, we added some information about the function of chloride anions of Pt-based drugs in Paragraph 2, Page 1, “Chloride anions improve the reduction potential and ability to receive electrons from some ligands like DAD, etc. However, the existence of chloride anions could probably improve the cytotoxicity of Pt-based drugs.”, according to the two documents the reviewer recommended.
Comment 2.3: I really liked the short section 2.2. about platinum nanoparticles. It is not entirely clear to me whether the examples presented were in vivo or in vitro? Only about one example was a comment about in vivo. If there were in vivo studies, I would very much like to look at the LD50 or LC50 data, and has the biodynamics of platinum nanoparticles been studied in living organisms?
Answer: Thanks for your kind reminder. Based on your suggestions, we added more details about platinum nanoparticles. The examples presented in section 2.2 about nanoparticles included experiments in vivo and in vitro. However, we choose not to provide the details of IC50 data related to nanoparticles presented in Figure 3 to make the manuscript brief. And it would be found in related documents such as “Carboplatin-Complexed and cRGD-Conjugated Unimolecular Nanoparticles for Targeted Ovarian Cancer Therapy” and so on. For example, in the related documents, such as the description of NANOPt-Pan in “Panitumumab-Conjugated Pt-Drug Nanomedicine for Enhanced Efficacy of Combination Targeted Chemotherapy against Colorectal Cancer”, characteristics of nanoparticles could be found in its Table 1.
Comment 2.4: Were the nanoparticles carcinogenic by any chance?
Answer: We really appreciated the kind reminder. We added more information about the toxicity of nanoparticle-based drug deliveries and provided some potential strategies to address these problems. We added the following part related to nanoparticle-based drug carriers' limitations in the right part of the revised version.
“Although great progress in nanoparticle-based drug deliveries has been reached in the past few decades, the toxicity and limitations should not be ignored. In nanoparticle-based drug carriers, liposomes are characteristic for self-assembling and assembling drugs to be liposome nanoparticles (LNP). However, liposomes tend to accumulate in the liver, spleen, and bone marrow, and organisms are also known as the mononuclear phagocytic system (MPS) in the human body. Liposomes accumulated in MPS could probably result in serious side effects and toxicity. To address these limitations, improving targeting ability and eliminating influences on other organisms could be the key. Louzhen Fan et al summarized the perspectives of nanomaterials such as liposomes, proteins, and carbon quantum dots as carriers, and also illustrates the progress of nanoparticle-based drugs in cancer medications.”
Comment 2.5: Page 11, “In addition to targeting DNA damage and mitochondrial damage in tumor cells, Jin-Lei Tian et al. [67] designed a series of dual-targeted monofunctional Pt(II) complexes 1-8 based on the aspect of improving drug delivery and targeting during antitumor therapy.” Compounds 1-8 came from the original paper – need to another numerate or delete the information.
Answer: We overlooked this detail and appreciate your kind and careful reminder. Page 11, We deleted the "1-8" in "complexes 1-8" in the revised version.

Reviewer 3 Report
Older but Stronger: Development of Platinum-based Anti-tumor Agents and Research Advances in Tumor Immunity
Manuscript ID: polymers - 2228720
Comments to the Authors
This paper reviews the development of several Platinum-based anti-tumor drugs in recent years and provides a good foundation for future research and development of novel Pt-based drugs. The authors summarizes the background of Pt-based drugs as anti-tumor agents, the related side-effects, and disadvantages. They have also reported a detailed overview of the current development in the field from chemical and immunological perspectives. The manuscript is rich in content and can prove to be very valuable for researchers in several fields such as, inorganic chemistry, anti-tumor drugs, and cancer among others.
The authors have provided a very well written manuscript with a good background on Pt-based anti-tumor agents, the related disadvantages, the currently adopted methodologies for improvements, and an outlook. They have also provided good technical details which will help as a good guideline to researchers in the field.
I have a few comments and suggestions below for some clarifications and better readability:
1. The authors have provided a good background on Pt-based anti-tumor drugs using cisplatin as the first example. In the introduction, they have explained the mechanism of cisplatin as anti-tumor drug works. It will be helpful if they can provide the structure of cisplatin as a figure and explain from their how the structure and chemistry works which they have explain pretty well.
2. The manuscript focus a lot on mitigating the adverse effects and disadvantages of Pt-based drugs. It might be useful if they provide more information on the different types of disadvantages discussed in the paper such as the different types of toxicity and what drug resistance actually means and why it happens.
3. The authors have presented nanocarriers with Pt-based drugs as being able to mitigate the adverse effects. Can they provide more explanations on how this happens? Additionally, nanoparticles are also known to be toxic to human bodies. How is their toxicity suppressed or addressed?
4. Can the authors also provide some information on if the nanocarriers with drugs have been applied commercially and how successful they have been?
5. Figures 2, 5, 8, and 9 have a lot of chemical structures which is useful but can they be presented in more organized manner for easy readability? I am wondering if they can be tabulated in some form with all the details present in the figure legends.

Author Response
Comment 1: This paper reviews the development of several Platinum-based anti-tumor drugs in recent years and provides a good foundation for future research and development of novel Pt-based drugs. The authors summarize the background of Pt-based drugs as anti-tumor agents, the related side effects, and the disadvantages. They have also reported a detailed overview of the current development in the field from chemical and immunological perspectives. The manuscript is rich in content and can prove to be very valuable for researchers in several fields such as inorganic chemistry, anti-tumor drugs, and cancer among others. The authors have provided a very well-written manuscript with a good background on Pt-based anti-tumor agents, the related disadvantages, the currently adopted methodologies for improvements, and an outlook. They have also provided good technical details which will help as a good guideline to researchers in the field.
Answer: We really cherished your kind assessments!
Comment 2: I have a few comments and suggestions below for some clarifications and better readability:
Answer: We really appreciated the kind reminder.
Comment 2.1. The authors have provided a good background on Pt-based anti-tumor drugs using cisplatin as the first example. In the introduction, they have explained the mechanism of cisplatin as anti-tumor drug works. It will be helpful if they can provide the structure of cisplatin as a figure and explain from their how the structure and chemistry work which they have explain pretty well.
Answer: Thanks for pointing this out. We restructured the layout of Figure 2, adding the structure of cisplatin and oxaliplatin as A and B in Figure 2, and explained the clinical situation of the extensively used anti-tumor drug, cisplatin, and the mechanism of how it works and the possible reasons for drug resistance related to low immune reaction, like an immune escape, etc.
As is shown on Page 1, right below Figure 2, we added some information and details about classical Pt-based drugs, cisplatin, and oxaliplatin as A and B in Figure 2.
As a result, in the Introduction part of the manuscript, we added “Generally speaking, Pt(IV) would be reduced into Pt(II) via the reductive microenvironment of Tumors. Compared to Pt(IV)-based anti-tumor drugs, Pt(II)-based ones perform better cytotoxic ability. Cisplatin, for example, has been used extensively in therapies for ovarian, prostate, testicular, lung, nasopharyngeal, and esophageal cancers, etc. through direct binding of DNA within tumor cells. However, this DNA binding is not targeting, causing damage to other organs, and resulting in limitation of clinical use.” for explanations of general mechanisms of Pt(IV)-, and Pt(II)-based medications.
Also, we added another example “Compared to cisplatin, oxaliplatin has moderate adverse effects and could be used for patients with hepatic dysfunction. And it is commonly used in the treatment of colorectal cancer.” to the Introduction part of the manuscript to further explain the limitations of Pt(II)-based medications.
Comment 2.2. The manuscript focuses a lot on mitigating the adverse effects and disadvantages of Pt-based drugs. It might be useful if they provide more information on the different types of disadvantages discussed in the paper such as the different types of toxicity and what drug resistance actually means and why it happens.
Answer: Thanks for the valuable comments and suggestions. We have completely rewritten this part, adding more information about the disadvantages of the different types, such as toxicity or drug resistance, and more explanation of why it happens.
Comment 2.3. The authors have presented nanocarriers with Pt-based drugs as being able to mitigate the adverse effects. Can they provide more explanations on how this happens? Additionally, nanoparticles are also known to be toxic to human bodies. How is their toxicity suppressed or addressed?
Answer: Thank you for your kind reminder. We surely overlooked the details about the nanoparticles' toxicity and the ways to address it. More details were displayed in the revised version.
As is shown on Page 2, “However, liposomes tend to accumulate in liver, spleen and bone marrow, and organisms are also known as the mononuclear phagocytic system (MPS) in the human body. Liposomes accumulated in MPS could probably result in serious side effects and toxicity.”
Comment 2.4. Can the authors also provide some information on if the nanocarriers with drugs have been applied commercially and how successful they have been?
Answer: Thanks for pointing this out. We considered the commercial use of nanocarriers in the revised version. However, we focused on the immune functions instead of their commercial usage. We take careful decisions not to add the commercial use of nanoparticle-based carriers.
Comment 2.5. Figures 2, 5, 8, and 9 have a lot of chemical structures which is useful but can they be presented in more organized manner for easy readability? I am wondering if they can be tabulated in some form with all the details present in the figure legends.
Answer: We reorganized Figures 2, 5, 8, and 9 according to the similarity of their structures and characteristics. As is shown in the following Figures.
